

# Emulating grid-based forest carbon dynamics using machine learning: an LPJ-GUESS v4.1.1 application

Carolina Natel[1], David Martín Belda[1], Peter Anthoni[1], Neele Haß[2], Sam Rabin[3] and Almut Arneth[1]

[1]Karlsruhe Institute of Technology, Institute of Meteorology and Climate Research/Atmospheric Environmental Research, Garmisch-Partenkirchen, Germany.
[2]Karlsruhe Institute of Technology, Institute of Geography and Geoecology, Karlsruhe, Germany
[3]National Center for Atmospheric Research, Climate and Global Dynamics, Boulder, United States

*Correspondence to*: Carolina Natel (carolina.moura@kit.edu)

**Abstract.** The assessment of forest-based climate change mitigation strategies relies on computationally intensive scenario analyses, particularly when dynamic vegetation models are coupled with socio-economic models in multi-model frameworks. In this study, we developed surrogate models for the LPJ-GUESS dynamic global vegetation model to accelerate the prediction of carbon stocks and fluxes, enabling quicker scenario optimization within a multi-model coupling framework. We trained two machine learning methods: random forest and neural network. We assessed and compared the emulators using performance metrics and Shapley-based explanations. Our emulation approach accurately captured global and biome-specific forest carbon dynamics, closely replicating the outputs of LPJ-GUESS for both historical (1850–2014) and future (2015–2100) periods under various climate scenarios. Among the two trained emulators, the neural network extrapolated better at the end of the century for carbon stocks and fluxes, and provided more physically consistent predictions, as verified by Shapley values. Overall, the emulators reduced the simulation execution time by 97%, bridging the gap between complex process-based models and the need for scalable and fast simulations. This offers a valuable tool for scenario analysis in the context of climate change mitigation, forest management, and policy development.

## 1 Introduction

Carbon sinks in natural and managed forests have become central elements of global climate change policy due to their relatively cost-effective climate change mitigation potential. Reduced deforestation, reforestation, agroforestry and improved forest management have an estimated mitigation potential ranging from 0.1 to 10.1 GtCO₂-equivalent per year, depending for example, on where they are implemented and the total area involved (Roe et al., 2019; Smith et al., 2020). In addition to these strategies, forest products such as wood can replace emission-intensive materials like steel and cement in construction while also storing carbon in the harvested wood (Churkina et al., 2020).

However, to investigate the potential of these practices, we must consider both the environmental changes that impact biogeochemical processes in forest ecosystems and the socioeconomic factors that affect land use and management. This



requires a sophisticated, multi-model approach. For example, the LandSyMM model (Alexander et al., 2023; Henry et al., 2022; Rabin et al., 2020) couples the LPJ-GUESS (Smith et al., 2014) dynamic global vegetation model (DGVM) with a land system and international trade model (PLUM) (Alexander et al., 2018). LPJ-GUESS simulates vegetation dynamics and biogeochemical cycles in response to different climate scenarios, while PLUM optimizes and projects future land use and

management based on socioeconomic scenario data and potential agricultural yields estimated by LPJ-GUESS (Alexander et al., 2023). Although this coupling has successfully modelled agricultural change scenarios and their effects on ecosystem services (Rabin et al., 2020), forest-based mitigation potential remains underexplored. A major barrier is the computational demand of simulating forest carbon dynamics. Forests, unlike crops, require long-term modeling—30 to 100 years—to account for growth, timber production, and carbon sequestration, which significantly increases the computational cost of optimization

within coupled frameworks.

To address computational efficiency issues, emulating process-based models (either in full or for specific components) has emerged as a powerful tool. Emulation involves building a simplified representation of a complex model by using input-output data from the original model simulations as training data (Franke et al., 2020). The resulting emulator is then capable of approximating the behavior of the original model with significantly reduced computation times. Emulators are particularly

well-suited to tasks such as rapid sensitivity analysis, model parameter calibration, and deriving confidence intervals (Reichstein et al., 2019). Over the years, several emulators have been developed and applied in environmental sciences, using techniques that range from simple linear and polynomial regressions (Ahlström et al., 2013; Ekholm et al., 2024; Franke et al., 2020) to more advanced techniques (Chen et al., 2018; Doury et al., 2023; Weber et al., 2020). Recently, machine learning (ML) algorithms have gained significant attention for their ability to accurately and efficiently model non-linear problems,

particularly in fields like earth system sciences, which often involve complex, high-dimensional datasets. For instance, ML-based emulators have been used to replace costly simulations of spatially resolved variables in large scale climate models (Beusch et al., 2020; Nath et al., 2021; Zhu et al., 2022), to support sensitivity analysis and efficient calibration of model parameters in land surface modeling (Dagon et al., 2020; Sawada, 2020), and to enable high-resolution simulations (Baker et al., 2022).

In spatially explicit, coupled socioeconomic models, decisions about a forestry-related land use taken today need to consider the potential return of the given forest (in terms of carbon storage or timber) several decades into the future. For these types of applications, the speedy runtime of a forest growth emulator is a significant advantage. This study aims to develop an emulator for LPJ-GUESS to enable faster optimizations within the LandSyMM model. We evaluated the accuracy and explainability of two ML methods, random forest (RF) and neural network (NN), and discussed potential limitations and future applications.

These methods were chosen for their ability to handle complex, non-linear relationships in ecological data and their proven performance in emulation tasks. The emulation approach presented here serves as a starting point, with future work aiming to explore emulations that incorporate various management interventions to better address questions related to forest-based climate change mitigation in the context of global environmental change.





## 2 Materials and Methods

### 2.1 LPJ-GUESS model

To train and evaluate the emulators, we used data generated by the LPJ-GUESS DGVM (Lindeskog et al., 2021; Smith et al., 2001, 2014). LPJ-GUESS simulates changes in vegetation composition and structure in response to atmospheric conditions such as climate and carbon dioxide concentration, nitrogen deposition, and land management from regional to global scales. The model represents natural vegetation as a mixture of co-occurring Plant Functional Types (PFTs). Vegetation dynamics are driven by stochastic gap dynamics, where the establishment, growth, and mortality of PFT age cohorts are modelled in a number of replicate patches for each simulated grid cell (Smith et al., 2014). Simulation of carbon dynamics is a key component of the model, including carbon uptake through photosynthesis, carbon allocation to plant tissues (leaves, roots, and wood), and carbon release through respiration and decomposition. The model has been extensively evaluated and has demonstrated its ability to capture large-scale vegetation patterns (Hickler et al., 2012) and the dynamics of the terrestrial carbon cycle (Lindeskog et al., 2021; Smith et al., 2014).

### 2.2 Emulation approach

The emulation process was formulated as a supervised learning regression problem, where the features $X_i$ (predictor variables) and targets $y_j$ (outputs) were derived from data generated by the process-based model. The modeling task was to predict either (a) carbon stocks (C stocks) in kgC m$^{-2}$, including vegetation carbon (VegC), soil carbon (SoilC), and litter carbon (LitterC), or (b) carbon fluxes (C fluxes) in kgC m$^{-2}$ year$^{-1}$, including gross primary productivity (GPP), net primary productivity (NPP), and heterotrophic respiration (Rh), for any given grid cell. We trained separate multi-output regressors for each prediction task (C stocks or C fluxes), as the performance of a single multi-task model was found to be inferior and is not presented here.

We selected 15 features, including variables related to climate, carbon states prior to a stand-replacing event, soil attributes, and a disturbance timer that tracks the time elapsed since the last stand-replacing disturbance (Table 1). These features were chosen based on preliminary analysis of the most important factors likely to influence annual forest C stocks and C fluxes in LPJ-GUESS.

**Table 1. Features and target variables**

| | Variable category | Variable description | Abbreviation | Unit | Temporal resolution |
|---|---|---|---|---|---|
| Features | Disturbance | Time elapsed since the last stand-replacing disturbance | time_since_disturbance | year | Annual |
| | Climate factors | Mean annual temperature | temp | °C | Annual |
| | | Total annual precipitation | prec | mm | Annual |
| | | Annual accumulated insolation | insol | W m$^{-2}$ | Annual |



|  |  | Minimum annual temperature | temp_min | °C | Annual |
|---|---|---|---|---|---|
|  |  | Maximum annual temperature | temp_max | °C | Annual / Annual |
|  |  | Highest mean monthly temperature | mtemp_max | °C | Annual |
|  |  | Total annual growing degree-days (accumulated sum on 0°C base in a year) | gdd0 | °C-day | Annual |
|  | Carbon state | Atmospheric $CO_2$ concentration | co2 | ppm | Constant |
|  |  | Initial (pre-disturbance) vegetation carbon pool | vegc_init | kgC m$^{-2}$ | Constant |
|  |  | Initial (pre-disturbance) litter carbon pool | litterc_init | kgC m$^{-2}$ | Constant |
|  |  | Initial (pre-disturbance) soil carbon pool | soilc_init | kgC m$^{-2}$ | Constant |
|  | Soil | Clay fraction | clay | % | Constant |
|  |  | Silt fraction | silt | % | Constant |
|  |  | Sand fraction | sand | % | Constant |
| Target variables | Carbon stocks (C stocks) | Vegetation carbon pool | VegC | kgC m$^{-2}$ | Annual |
|  |  | Soil carbon pool | SoilC | kgC m$^{-2}$ | Annual |
|  |  | Litter carbon pool | LitterC | kgC m$^{-2}$ | Annual |
|  | Carbon fluxes (C fluxes) | Gross primary productivity | GPP | kgC m$^{-2}$ yr$^{-1}$ | Annual |
|  |  | Net primary productivity | NPP | kgC m$^{-2}$ yr$^{-1}$ | Annual |
|  |  | Heterotrophic respiration | Rh | kgC m$^{-2}$ yr$^{-1}$ | Annual |

### 2.2.1 Random Forest

We developed Random Forest (RF) regressors (Breiman, 2001) using the scikit-learn Python library's implementation of the RandomForestRegressor (Pedregosa et al., 2011). Mean squared error (MSE) was used as the criterion for evaluating the quality of a decision split. The RF model employed a bootstrap strategy, where training samples were drawn with replacement to fit each tree. A fixed random seed (seed = 42) was used to ensure reproducibility. RF selected hyperparameters were optimized via grid search, with the selected values and final best settings presented in Table S1 of the Supporting Information.



### 2.2.2 Neural Network

We developed neural network (NN) regressors using the TensorFlow and Keras libraries (Chollet, 2015, TensorFlow developers, 2024), constructing a fully connected feedforward neural network. The architecture included an input layer corresponding to the feature space, followed by a series of hidden layers characterized by the number of neurons, activation function, and dropout rate. NN-selected hyperparameters were optimized via grid search, with the selected values and final best settings presented in Table S2 of the Supporting Information.

The output layer consisted of three distinct nodes, each corresponding to one of the target variables (either VegC, SoilC and LitterC for the C stocks regressor or GPP, NPP and Rh for the C fluxes regressor). Each output node employed a linear activation function, producing a scalar value for the respective C stock or C flux. The model was compiled using the Adam optimizer (Kingma and Ba, 2017), and the MSE as the loss function. The model was trained for up to 1000 epochs, with an early stopping callback monitoring the validation loss with a patience of 10 epochs. The training process was stopped early when no improvement in validation loss was observed, and the best weights were restored. We used a seed number (seed=42) to initialize the weights and bias and ensure reproducibility of our results.

The input features were normalized using the $MinMax$ scaler from the scikit-learn library (Equation 1) to ensure all variables were on a comparable scale, thereby accelerating convergence during training.

$$X_{scaled} = \frac{X - X_{min}}{X_{max} - X_{min}}, \tag{1}$$

where $X_{scaled}$ is the normalized value, $X$ is the original value, $X_{min}$ is the minimum value of the feature, $X_{max}$ is the maximum value of the feature.

For NN predictions, we applied a post-processing step to enforce non-negativity in predicted C stocks by replacing all negative values with zero, as negative predictions are not meaningful in this context. This step was not necessary for RFs, which naturally avoid producing negative values due to their structure. Predictions for C fluxes were not post-processed, as they can accept negative values representing flux to the ecosystem in LPJ-GUESS. This approach ensures that all predictions remain physically interpretable.

### 2.3 Evaluation

The performance of the emulator was evaluated using three metrics: normalized root mean square error (NRMSE), relative bias, and the coefficient of determination ($R^2$). These metrics were computed for each target variable to assess the emulators' predictive accuracy. The NRMSE (Equation 2) is a normalized version of the root mean square error, scaled by the range of the observed values. The relative bias (Equation 3) quantifies the systematic error between the predicted and true values as a percentage, and the $R^2$ (Equation 4) indicates the proportion of variance in the true values explained by the emulators.

$$NRMSE = \frac{\sqrt{\sum_{i=1}^{n}(y_i - \hat{y}_i)^2}}{max(y_i) - min(y_i)}, \tag{2}$$





$$Relative\ bias = \frac{\sum_{i=1}^{n} \hat{y}_i - y_i}{\sum_{i=1}^{n} y_i} . 100, \tag{3}$$

$$R^2 = 1 - \frac{\sum_{i=1}^{n}(y|i - \hat{y}_i)^2}{\sum_{i=1}^{n}(y|i - \overline{y}_i)^2} \tag{4}$$

In these equations, $y_i$ represents the "true" values as simulated by LPJ-GUESS, $\overline{y}_i$ is the average of all the "true" values $y_i$, $\hat{y}_i$ denotes the predicted values from the emulator, and $n$ is the number of observations.

The NRMSE provides a normalized error magnitude, enabling comparison across different target variables. Relative bias offers insight into systematic deviations between predictions and true values, while R² indicates the goodness of fit. To evaluate the spatial generalization of the emulator, these metrics were calculated on a test set of grid cells not used during training or validation (see Section 3.2). In addition, the emulator's ability to extrapolate was tested by applying it to climate scenarios not included in the training or validation phases.

**2.4 Explainable machine learning**

Understanding ML model predictions is essential for evaluating their reliability and gaining insights into the factors driving the predictions. In this study, we complemented the evaluation of our models by incorporating SHAP (SHapley Additive Explanations) values analysis. SHAP is a method based on cooperative game theory used to increase transparency and interpretability of ML models, available in the SHAP Python library (Lundberg and Lee, 2017). SHAP values indicate the 135 most influential features and the direction in which changes in feature values may affect the predicted output. This technique was chosen due to its model-agnostic nature, allowing for consistent interpretability across different algorithms. To reduce computational time, the SHAP analysis was conducted on a randomly sampled subset of the test dataset (n=500) from the predictions made using the MPI-ESM1-2-HR forcing data for the historical period and climate scenarios (RCP2.6, RCP4.5, RCP7.0 and RCP8.5).

**2.5 Computational gain**

To evaluate the computational efficiency of our emulators, we compared the execution time of the RF and NN models against the original LPJ-GUESS model. The timing for the emulators encompassed model and data loading, and prediction phases for both carbon stocks and fluxes. We quantified the computational efficiency gain using Equation 5.

$$gain = \frac{t_{LPJ-GUESS} - (t_{cstocks} + t_{cfluxes})}{t_{LPJ-GUESS}} 100, \tag{5}$$

where $gain$ is the computational gain, $t_{LPJ-GUESS}$ is the baseline execution time (LPJ-GUESS model), $t_{cstocks}$ is the execution time of the emulator for carbon stocks' predictions, and $t_{cfluxes}$ is the execution time of the emulator for carbon fluxes' predictions. We calculated this efficiency metric separately for both the RF and NN models. It's important to note that while



LPJ-GUESS simulates all output variables simultaneously, our emulation approach requires separate models for C fluxes and C stocks. Therefore, we summed the execution times of both task-specific emulators to ensure a fair comparison with the LPJ-150 GUESS runtime.

LPJ-GUESS's computational demand scales quasi-linearly with the number of simulated model grid cells and years, as each pixel is processed independently. For our benchmark, we estimated the computational gain using predictions for a 165-year historical period (1850–2015) simulated with the MPI-ESM1-2-HR climate model. While this period was selected for practical reasons in our calculations, it is also a widely used simulation period in climate modeling studies. We used a single grid cell 155 (0.5° x 0.5°) for this comparison, noting that the gain would scale proportionally for larger areas. For the LPJ-GUESS simulation, we excluded the spin-up period from the timing, instead initializing the historical simulations from a pre-computed state file.

We excluded the time required for generating the training data, training and evaluating the emulators, and the initial development of the LPJ-GUESS model, as these steps occur only during the development phase and are not part of the 160 emulators'operational use. Our focus was on comparing the runtime efficiency of the trained emulators against LPJ-GUESS for making predictions, which reflects their typical practical application. It should be noted that the actual time savings depend on the machine infrastructure and software, and may therefore differ from the theoretical estimate.

## 3 Data

### 3.1 Data generation

We conducted multiple scenario simulations using the LPJ-GUESS model, focusing specifically on forest grid cells. These cells were identified through an initial global simulation of potential natural vegetation, followed by classification into distinct vegetation types (biomes) based on PFT abundances and leaf area index, as described by Smith et al. (2014). From the globally simulated forest grid cells, we employed a stratified sampling method to ensure balanced representation across eight distinct forest biomes: tropical rainforest, tropical deciduous forest, tropical seasonal forest, boreal evergreen forest, boreal deciduous 170 forest, temperate broad-leaved evergreen forest, temperate deciduous forest, and temperate and boreal mixed forests. A total of 15% of the forest grid cells were selected for emulator development, resulting in a dataset of 3,448 grid cells.

The sampled grid cells were then randomly divided into training (80%), validation (10%), and test (10%) sets, with an equal number of cells selected from each biome for each set (Fig. 1). This approach minimizes the risk of over-representing any particular forest biome during training and evaluation, thereby reducing bias in the ML models (Sun et al., 2023), as biophysical 175 properties and climate change responses can vary significantly between them. The training set was used to update the ML model parameters, while the validation set guided hyperparameter tuning and monitoring for overfitting. The test set was reserved for evaluating emulation performance on grid cells unseen during training and validation. By using distinct grid cells for training, validation, and testing, we aimed to assess the robustness of the spatial generalization of the emulation.



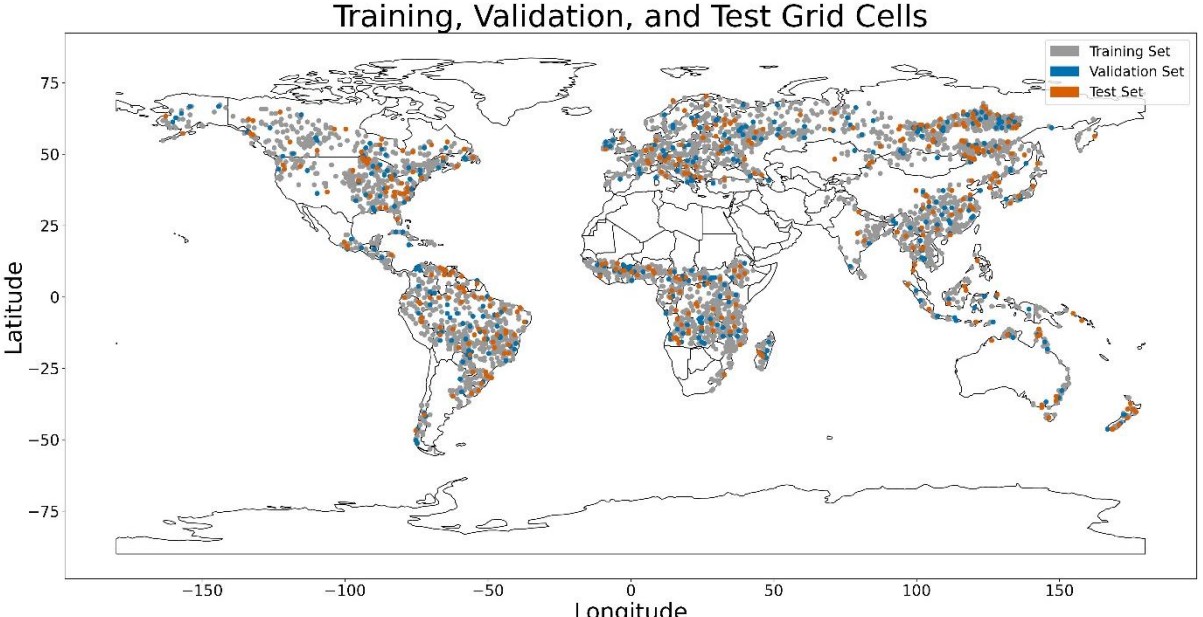

**Figure 1: Spatial distribution of the sampled grid cells for training, validation and test.**

The LPJ-GUESS simulations were driven by climate scenarios derived from the Coupled Model Intercomparison Project Phase 6 (CMIP6; Eyring et al., 2016), bias-corrected for the Inter-Sectoral Impact Model Intercomparison Project phase 3 (ISIMIP3) (Lange, 2019; Lange and Büchner, 2021). We used temperature, precipitation, and solar radiation data from five Earth System Models (ESMs): IPSL-CM6A-LR, MPI-ESM1-2-HR, MRI-ESM2-0, GFDL-ESM4, and UKESM1-0-LL, covering four Representative Concentration Pathways (RCPs): RCP2.6, RCP4.5, RCP7.0, and RCP8.5. Nitrogen deposition data used in the simulations was sourced from Lamarque et al., 2013. Details of the LPJ-GUESS simulation protocol, including model version, modifications, experimental setup, and forcing data, are available in the Code and data availability section and Supporting Information S1.

### 3.2 Data preprocessing

To make the emulator model agnostic, we pre-processed the raw LPJ-GUESS simulation outputs by taking the ensemble mean of the five ESMs for training. The test set was used to assess the emulation performance on data not used during training and validation. For testing, we used the LPJ-GUESS simulation outputs from each climate model individually, rather than the ensemble mean, and included climate change scenarios not used during training and validation. The data sets are described in Table 2.





**Table 2. Description of the training, validation and test datasets. The datasets covered the historical (1850 - 2014) and future period (2015 - 2100).**

| Data set | Number of grid cells | Number of samples | Climate |
|---|---|---|---|
| Training | 2760 | 930120 | Historical and future projections under RCP2.6 and RCP8.5 |
| Validation | 344 | 115928 | Historical and future projections under RCP2.6 and RCP8.5 |
| Test | 344 | 411768 | Historical and future projections under RCP2.6, RCP4.5, RCP7.0 and RCP8.5 |

The number of samples refers to the total number of data points in each dataset, calculated by multiplying the number of grid cells by the number of time points (years) in the climate data for each set. Historical data contains 165 years, and the future period contains 86 years. The training and validation sets use ensemble data from five climate models. The test set uses individual simulations from three separate climate models (GFDL-ESM4, MPI-ESM1-2-HR and MRI-ESM2-0) rather than an ensemble, so we multiplied the future period length by three.

## 4 Results

### 4.1 Emulator performance

The emulators demonstrated a significant reduction in simulation execution time, with a 97% decrease observed when compared to the execution time of LPJ-GUESS. We evaluated the emulators' ability to replicate carbon dynamics as simulated by LPJ-GUESS using the test dataset. The emulators were trained on historical climate data and projections from the RCP2.6

and RCP8.5 scenarios. Performance was assessed over the historical period and across four RCP scenarios—RCP2.6, RCP4.5, RCP7.0, and RCP8.5—capturing a broad range of potential future climate conditions.

### 4.1.1 Carbon stocks

Overall, both the NN and RF models demonstrated good performance across the different carbon pools and RCP scenarios. As shown in Table 3, the emulators were able to generalize to LPJ-GUESS outputs produced with climate projections not

included in the training data without a significant decline in performance. The NRMSE values were consistently low, ranging from 0.01 to 0.12 across target variables and scenarios, indicating a high degree of predictive accuracy. The RF underestimated C stocks for most of the RCP scenarios. Overall, the NN model exhibited consistently smaller relative bias compared to the RF model, especially in the prediction of VegC, and except for SoilC. Among the carbon stock variables, SoilC was predicted with the greatest accuracy, exhibiting the lowest error and highest R² values. Overall, the NN emulator outperformed the RF

emulator across the RCP scenarios and target variables.



**Table 3. Performance metrics of Random Forest (RF) and Neural Network (NN) emulators for predicting carbon stocks in forest ecosystems, including vegetation carbon (VegC), soil carbon (SoilC), and litter carbon (LitterC), across four different climate projections.**

| | | VegC | | SoilC | | LitterC | |
|---|---|---|---|---|---|---|---|
| | | NN | RF | NN | RF | NN | RF |
| | Historical | 0.09 | 0.08 | 0.02 | 0.02 | 0.06 | 0.05 |
| | RCP2.6 | 0.12 | 0.11 | 0.02 | 0.02 | 0.09 | 0.09 |
| NRMSE | RCP4.5 | 0.11 | 0.11 | 0.02 | 0.03 | 0.09 | 0.09 |
| | RCP7.0 | 0.11 | 0.1 | 0.02 | 0.03 | 0.09 | 0.09 |
| | RCP8.5 | 0.10 | 0.1 | 0.03 | 0.03 | 0.09 | 0.09 |
| | Historical | 1.4 | 5.58 | 1.63 | 0.32 | 3.16 | 2.92 |
| | RCP2.6 | 1.6 | -1.91 | 1.17 | -0.31 | -2.45 | -2.69 |
| Relative bias (%) | RCP4.5 | -0.18 | -4.75 | 0.88 | -0.47 | -2.16 | -3.34 |
| | RCP7.0 | 0.24 | -4.65 | 0.58 | -0.37 | -0.64 | -2.17 |
| | RCP8.5 | 0.3 | -5.55 | -0.08 | -0.42 | -0.08 | -2.33 |
| | Historical | 0.79 | 0.81 | 0.99 | 0.98 | 0.76 | 0.85 |
| | RCP2.6 | 0.52 | 0.59 | 0.98 | 0.98 | 0.72 | 0.76 |
| $R^2$ | RCP4.5 | 0.54 | 0.58 | 0.99 | 0.98 | 0.73 | 0.75 |
| | RCP7.0 | 0.57 | 0.60 | 0.99 | 0.98 | 0.72 | 0.75 |
| | RCP8.5 | 0.62 | 0.61 | 0.98 | 0.98 | 0.73 | 0.76 |

Figure 2 illustrates how well the emulators captured the temporal dynamics of carbon pools across different biomes after a stand-replacing disturbance. Overall, both emulators closely approximated the process-based simulations from historical

period into future projections across RCPs. However, in boreal forests, the RF regressor tends to overestimate VegC during the early years of forest regrowth following a disturbance, while underestimating it by the end of the 21st century across all RCPs. Similarly, the NN regressor showed this pattern in tropical forests, where it also struggled to capture the undulations in VegC likely associated with tree age-related mortality. In temperate and mixed forests, both emulators accurately represented initial regrowth, but failed to capture VegC accurately by the end of the 21st century.

Both emulators underestimated the peak in LitterC following a disturbance and toward the end of the time series, particularly in temperate and mixed forests. SoilC remains relatively stable over time, with the RF emulator capturing these subtle changes more effectively than the NN emulator, particularly in boreal and tropical forests.



**Figure 2: Biome-specific average carbon stocks trajectories across a range of climate change projections following a stand-replacing disturbance in 1849. Solid lines represent LPJ-GUESS simulations, while dashed and dotted lines indicate emulators' predictions. The predictions are averaged across three Earth System Models (GFDL-ESM4, MPI-ESM1-2-HR, MRI-ESM2-0) for the test set.**

Figure 3 presents the spatial patterns of emulator errors in carbon stock predictions. The RF and NN emulators exhibited distinct spatial errors for VegC. The RF generally underestimated VegC, except for significant overestimation in Central Asia. In contrast, the NN tends to overestimate VegC in boreal evergreen forests in Central Asia and in temperate broad-leaved evergreen forests, particularly in Eastern Asia, the southeastern United States, and Southern Europe. Additionally, the NN overestimated VegC at the borders of forest biomes, including boreal forests in Russia and North America. Both emulators demonstrated high accuracy in SoilC predictions, with minimal spatial error variation across regions. For LitterC, the RF and NN models showed similar error patterns in tropical forests, although the NN model tends to produce higher errors, especially in Eastern China.



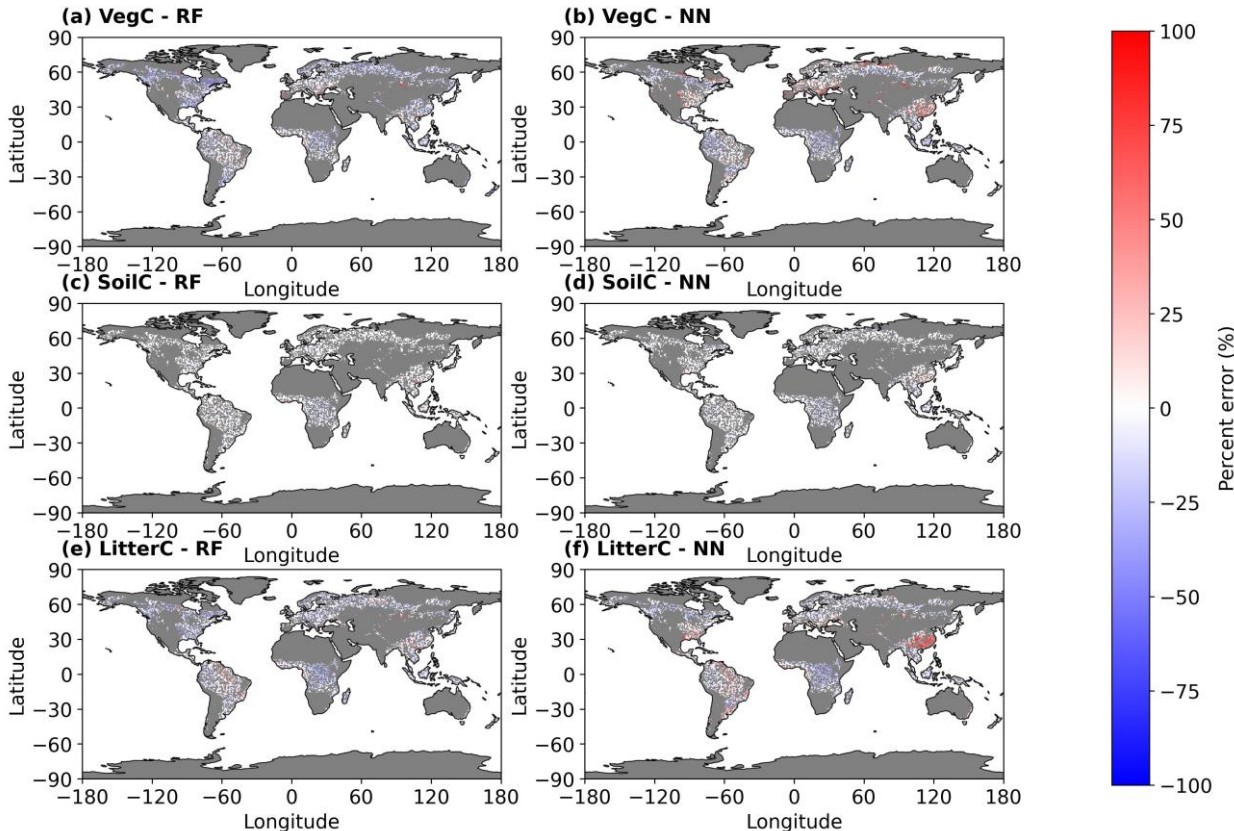

**Figure 3: Average percent error in vegetation carbon (VegC), soil carbon (SoilC), and litter carbon (LitterC) stocks (kgC m⁻²) predicted by Random Forest (RF) and Neural Network (NN) emulators, compared to LPJ-GUESS model outputs. Errors are averaged over the period 2070–2100. The predictions are based on simulations using climate data from the MPI-ESM1-2-HR Earth System Model under the RCP8.5 scenario. The map illustrates predicted errors across all LPJ-GUESS forested grid cells, including those used for training, validation, and test. Grey pixels indicate non-forested areas.**

### 4.1.2 Carbon fluxes

Similar to the C stocks, the NN and RF emulators showed good agreement with the LPJ-GUESS simulations in predicting GPP, NPP, and Rh under the historical period and four RCPs. Both emulators successfully generalized to RCPs not used during the training process. The NRMSE values were consistently low, ranging from 0.07 to 0.10 for both emulators across all carbon flux variables and RCP scenarios, indicating accurate predictions (Table 4).

Overall, the RF model outperformed the NN during the historical period, exhibiting lower error and more accurate predictions. However, performance differences between the models became less pronounced for the RCP scenarios. In the warmer RCP scenarios, the NN showed a slight improvement over the RF model, with the exception of Rh, where the RF model maintained a marginal advantage. This suggests that while the RF model excels under historical conditions, the NN may adapt better to projected warmer climates, providing competitive performance across most C fluxes.





**Table 4. Performance metrics of Random Forest (RF) and Neural Network (NN) emulators for predicting carbon fluxes in forest ecosystems, including gross primary productivity (GPP), net primary productivity (NPP), and heterotrophic respiration, (Rh) across four different climate projections.**

|  |  | GPP | | NPP | | Rh | |
|---|---|---|---|---|---|---|---|
|  |  | NN | RF | NN | RF | NN | RF |
| NRMSE | Historical | 0.07 | 0.07 | 0.08 | 0.08 | 0.07 | 0.07 |
|  | RCP2.6 | 0.09 | 0.10 | 0.08 | 0.09 | 0.10 | 0.10 |
|  | RCP4.5 | 0.09 | 0.10 | 0.08 | 0.09 | 0.10 | 0.10 |
|  | RCP7.0 | 0.08 | 0.09 | 0.08 | 0.09 | 0.09 | 0.10 |
|  | RCP8.5 | 0.07 | 0.08 | 0.08 | 0.08 | 0.09 | 0.10 |
| Relative bias (%) | Historical | 4.62 | 0.07 | 3.2 | 0.17 | 2.87 | 0.1 |
|  | RCP2.6 | 0.98 | -0.45 | -0.19 | -0.68 | 1.45 | 0.11 |
|  | RCP4.5 | 1.02 | 2.99 | -0.36 | 1.93 | 1.55 | 2.6 |
|  | RCP7.0 | 0.65 | 1.53 | -0.25 | 0.8 | 1.88 | 2.14 |
|  | RCP8.5 | -0.33 | -1.61 | -1.28 | -2.37 | 2.1 | 0.32 |
| $R^2$ | Historical | 0.68 | 0.72 | 0.62 | 0.61 | 0.67 | 0.67 |
|  | RCP2.6 | 0.61 | 0.58 | 0.51 | 0.46 | 0.51 | 0.52 |
|  | RCP4.5 | 0.66 | 0.51 | 0.54 | 0.42 | 0.52 | 0.48 |
|  | RCP7.0 | 0.73 | 0.66 | 0.57 | 0.48 | 0.55 | 0.52 |
|  | RCP8.5 | 0.8 | 0.73 | 0.63 | 0.53 | 0.6 | 0.57 |

The emulators reasonably captured the temporal dynamics of C fluxes across different biomes following a stand-replacing
disturbance, as shown in Fig. 4. However, some discrepancies were observed with the RF emulator's predictions of GPP and
NPP toward the end of the century across various RCPs and forest types. NN emulator systematically overestimated GPP in
boreal forests. Figure 5 shows the spatial variation in the emulators' errors for C fluxes. The average error for the period 2070–
2100 was small, and the error patterns were quite similar between the emulators.



**Figure 4: Biome-specific average carbon fluxes across four climate projections (columns) following a stand-replacing disturbance at year 1849. Solid lines indicate LPJ-GUESS simulations, and dashed lines indicate emulators. The predictions are averaged across three Earth System Models (GFDL-ESM4, MPI-ESM1-2-HR, MRI-ESM2-0) for the test set.**





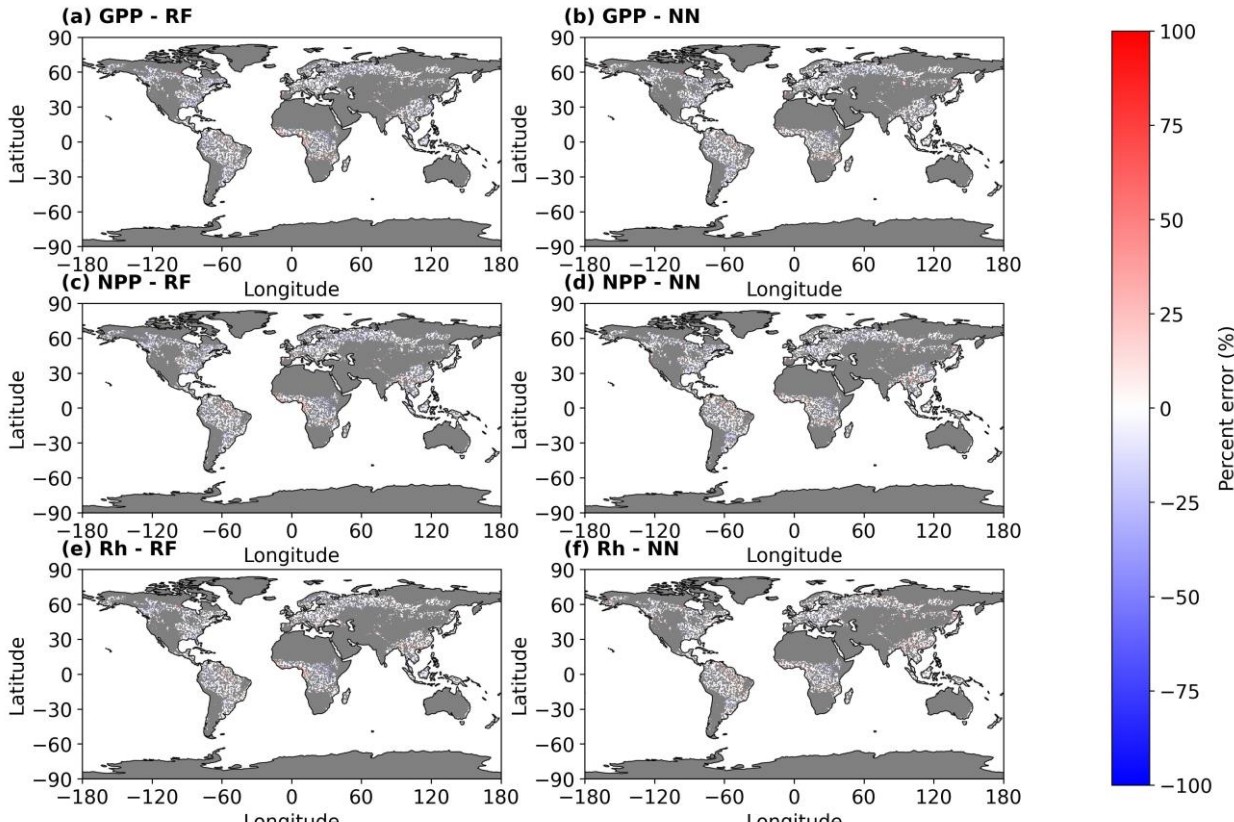

**Figure 5: Average percent error in gross primary productivity (GPP), net primary productivity (NPP), and heterotrophic respiration (Rh) (kgC m⁻² yr⁻¹) predicted by Random Forest (RF) and Neural Network (NN) emulators, compared to LPJ-GUESS model outputs. Errors are averaged over the period 2070–2100. The predictions are based on simulations using climate data from the MPI-ESM1-2-HR Earth System Model under the RCP8.5 scenario. The map illustrates predicted errors across all LPJ-GUESS forested grid cells, including those used for training, validation, and test. Grey pixels indicate non-forested areas.**

## 4.2 Emulator explainability

We extended our emulation assessment by presenting an analysis of model explainability through SHAP values. This technique provided complementary insights into the inner workings of our ML-based emulators, indicating which features were most influential in driving predictions and how they impacted the final output.

### 4.2.1 Carbon stocks

The RF and NN emulators exhibited varying feature importance and feature value impacts on carbon stock predictions (Fig. 6). For VegC, the NN model ranked pre-disturbance VegC pool, growing degree-days, and annual mean temperature as the most influential features, while the RF model ranked initial VegC, time since disturbance, and $CO_2$. There was a discrepancy between the models in terms of the impact of annual mean temperature. While the NN model attributed a positive effect of lower temperatures on VegC, the RF model indicated a decrease in VegC under lower temperature values. Both models





indicated a positive effect of the time since disturbance on VegC, which is consistent with the expected regrowth of vegetation
carbon following a disturbance.

For SoilC, the NN model is primarily influenced by the initial SoilC state, temperature, and soil properties (sand, silt, and clay
fractions), while the RF model relied mainly on the initial SoilC state. In NN model predictions, higher temperatures lead to
an increase in soil carbon content. This result may seem counterintuitive, as higher temperatures usually increase Rh, reducing
SoilC. However, in the warmer RCP scenarios where LPJ-GUESS simulates soil carbon increases, such as in tropical forests,
NPP rises faster than Rh (Figs. 2 and 4). This might lead to a net carbon gain, suggesting that biomass input to the soil pool
exceeds the increase in Rh.

For LitterC predictions, the NN model identified the initial LitterC state, growing degree-days, and soil sand fraction as the
most significant features. In contrast, the RF model highlighted the initial carbon states of vegetation, soil, and litter, followed
by time since disturbance and atmospheric $CO_2$ concentration, as more important. Both models concurred that in the initial
305    years following a disturbance, lower time-since-disturbance values were associated with higher LitterC, which aligns with the
observed peak in this pool after disturbance events.

Overall, variables associated with the carbon history of the stand before the disturbance are highly ranked in feature
importance. Additionally, the time since disturbance, atmospheric $CO_2$ concentration, growing degree-days, and mean annual
temperature also appeared to be crucial for predicting C stocks.





**Figure 6: SHAP values per feature for carbon stock predictions (vegetation carbon (VegC), soil carbon (SoilC), and litter carbon (LitterC)) using the (a - c) Random Forest (RF) and (d - f) Neural Network (NN) emulators. The Y-axis lists the features used in the model. The X-axis displays SHAP values, which quantify the impact of each feature on the model's prediction. Positive SHAP values indicate that a feature increases the prediction, while negative SHAP values suggest a decrease. The color gradient represents the feature values (red for high, blue for low). Each point on the plot corresponds to a single data point from the dataset, and its position along the X-axis shows the contribution of that feature to the prediction for that instance. For example, in 6a, low values (blue) of time elapsed since last disturbance (time_since_disturbance) decrease the predicted VegC, while high values (red) increase the predicted VegC by up to 2.**

### 4.2.2 Carbon fluxes

The SHAP analysis of C fluxes revealed contrasting patterns of feature importance and feature value impacts between the RF and NN models (Fig. 7). In the RF model, initial vegetation carbon state, atmospheric $CO_2$ concentration, and minimum



temperature consistently emerged as the most influential features across all carbon flux predictions. In contrast, the NN model identified growing degree-days above 0°C, atmospheric $CO_2$ concentration, initial vegetation carbon, and highest mean

monthly temperature as the primary drivers.



**Figure 7: SHAP values per feature for carbon flux predictions (gross primary productivity (GPP), net primary productivity (NPP), and heterotrophic respiration (Rh)) using the (a - c) Random Forest (RF) and (d - f) Neural Network (NN) emulators. For a detailed explanation of the SHAP plot, refer to the caption of Fig. 6.**



## 5 Discussion

We developed a ML emulation approach to approximate LPJ-GUESS simulations of carbon dynamics in forest ecosystems. Both RF and NN emulators accurately reproduced the process-based output. However, differences in their generalization to climate scenarios and ML explainability revealed distinct strengths and weaknesses inherent to each model type.

### 5.1 Emulator performance

Overall, the RF emulator exhibited lower bias during the historical period and outperformed the NN in predicting Rh and in capturing small changes in SoilC. The NN model, on the other hand, outperformed the RF emulator in predicting VegC and under the warmer RCP scenarios. It also showed superior performance for Rh predictions in mixed forests toward the end of the century. The generalization power of any learning algorithm depends on the inductive biases of that algorithm (Mitchell and Sheppard, 2019). RF models, for example, partition data through decision node splits using step functions that approximate relationships between inputs and outputs (Breiman et al., 2017). While this structure works well within the bounds of the training data, it limits the model's capacity to smoothly extrapolate outside the training data distribution. In contrast, NNs excel at modeling continuous relationships, making them more capable of generalizing to unseen data, particularly when extrapolation is required (Muckley et al., 2023). Given that our dataset contains less data for warmer climates near the end of the century, this limitation may affect the RF model's performance, particularly in extrapolating to these conditions. However, NNs are generally more robust in handling such extrapolation tasks due to their smoother function approximations. This suggests that RF may be more reliable when future projections remain close to the training data distribution or for interpolation tasks, while NNs could be better suited for scenarios where extrapolation is critical. This distinction is particularly relevant for climate assessments, where projecting across a wide range of future scenarios is essential for decision-making. Other studies comparing these two ML models have also found negligible differences in their performance (Ahmad et al., 2017), with some reporting marginal improvements by RF in regression tasks, especially in tabular data (Nawar and Mouazen, 2017, Grinsztajn et al., 2022). The performance of ML algorithms varies significantly depending on the dataset's dimensionality and the specific application. NNs generally require larger datasets to achieve optimal performance, while RFs are more data-efficient, needing fewer samples for training and minimal hyperparameter tuning. Therefore, the choice of algorithm should always be evaluated in the context of the specific application at hand.

It should also be noted that training emulators on climate projections from ESMs was not our initial approach. We first experimented with stylized climate change scenarios for training, following the methodology of Franke et al., 2020 using statistical methods for the emulation of crop productivities. Our assumption was that ML algorithms could learn generalized relationships between inputs and outputs and subsequently generalize well to realistic climate projections. However, the trained models failed to extrapolate effectively to the real CMIP6 climate scenarios. A plausible explanation for this is that the stylized scenarios produced combinations of temperature, precipitation, and atmospheric $CO_2$ concentrations too far removed from those expected in realistic settings. This could have biased the models, reducing their effectiveness when faced with actual





climate projections. This reinforces the importance of training ML models on data that closely mirrors the target problem. Furthermore, it is important to consider the differences between crops and forest ecosystems. In crops, factors such as time since disturbance typically play a less direct role, as the annual cycle of harvesting and replanting limits the long-term effects of disturbances. Crops are more strongly influenced by climate and $CO_2$ levels. In contrast, trees are significantly affected by legacy effects and timing of events, which play a crucial role in carbon dynamics. This disparity highlights the need for emulators to incorporate these variables when applied to forest ecosystems.

## 5.2 Emulator explainability

The selection of predictive models in scientific modeling often overlooks explainability, especially when high predictive accuracy is a primary objective. However, explaining ML models is important for diagnosing model biases, managing multi-objective trade-offs, and mitigating unexpected outcomes in practical applications (Muckley et al., 2023). In this study, the most important features varied across models and prediction tasks, reflecting how different ML approaches prioritize distinct aspects of the input data.

### 5.2.1 Carbon stocks

In regard to VegC, both models concur that the initial state of VegC is the most significant predictor. However, they diverge on other influential factors. The NN model emphasizes growing degree-days and annual mean temperature, whereas the RF model emphasizes time since disturbance and $CO_2$ concentration. This suggests that the RF model is more sensitive to disturbance history and $CO_2$ levels, whereas the NN model captures temperature-related dynamics more strongly.

Both models indicate that the time elapsed since a disturbance positively impacts VegC, suggesting that the emulators effectively capture the dynamics of post-disturbance vegetation recovery. The rate of carbon accumulation is significantly influenced by the age of the forest stand. Younger forests, which have recently experienced disturbances, tend to absorb carbon at a much faster rate than older forests. In mature forests, carbon accumulation slows as trees approach their maximum growth potential (Cook-Patton et al., 2020; Pugh et al., 2019). Additionally, lower values of time since disturbance, representing the initial years following a disturbance, are associated with higher levels of LitterC. This feature helps capture the observed peak in the litter carbon pool from biomass killed during the disturbance (Zhang et al., 2024).

Initial carbon pool states were also important features, playing a central role in predicting both soil carbon and vegetation carbon pools. The initial VegC state might indicate how much carbon from organic matter is transferred to the litter and soil pools after a disturbance. Meanwhile, the initial SoilC pool may reflect the forest's carbon carrying capacity under prevailing environmental conditions. Without considering spatial proxies like coordinates in the emulation, these features may help differentiate biome-specific carbon dynamics. For instance, tropical forests store large amounts of carbon in aboveground biomass, while boreal forests store more carbon underground. We suppose that this bioclimatic variation is captured by the initial carbon pool features, offering insights into the potential carbon saturation of different ecosystems.




In the case of predicting SoilC, both emulators demonstrated excellent performance, with low errors and R² higher than 0.98. However, this may represent an overly optimistic result, partly attributed to the inclusion of the highly correlated initial SoilC
pool as a feature. Since SoilC exhibits relatively small temporal variation, the inclusion of this feature might have exaggerated the model's performance by making the prediction task less challenging. The NN model incorporates a broader range of factors, such as temperature and soil attributes (sand, silt, clay fractions), suggesting it accounts for more complex interactions in soil dynamics. However, the RF model relies primarily on the initial SoilC state, implying that it gives less weight to environmental variables and might perform more conservatively in predicting soil carbon changes over time. That said, such a general
prediction is beyond the scope of our stated objective, and we consider it legitimate to use this kind of information in surrogate models to speed up calculations needed for assessments. Nevertheless, to avoid this over-reliance on initial soil carbon state, future iterations of the emulator could apply more advanced regularization techniques to mitigate its influence in the overall output.

For LitterC, the NN model emphasized initial LitterC, gdd0, soil sand fraction, and annual mean temperature, reflecting
sensitivity to regional soil conditions and environmental factors. In contrast, the RF model focused on initial carbon pool states (VegC, SoilC, and LitterC), disturbance history, and $CO_2$ concentration, indicating greater reliance on initial carbon conditions and disturbance-driven dynamics. The RF model seems to capture the indirect impact of atmospheric $CO_2$ on photosynthetic activity, which drives vegetation growth and ultimately influences litter carbon through increased biomass transfer to the litter pool.

Overall, the NN model appears to capture more complex ecological relationships, especially involving temperature and soil characteristics, which may make it better suited for understanding nuanced ecosystem processes, the RF model offers a more straightforward interpretation centered on disturbance and initial conditions. However, it tends to produce more conservative predictions and may overlook certain climatic variations across scenarios. Although ML explainability does not reveal the exact predictive value of each feature, it provides valuable insights into how individual features influence model behavior.

**5.2.2 Carbon fluxes**

Both models effectively capture the dominant role of atmospheric $CO_2$ in photosynthesis and the critical influence of initial vegetation carbon on potential carbon uptake and release. However, they diverge in their treatment of temperature variables. The RF model places greater emphasis on minimum temperature, suggesting a focus on colder temperature thresholds. In contrast, the NN model prioritizes growing degree-days and maximum mean temperature, indicating a more complex
representation of how temperature extremes and accumulated warmth affect both photosynthesis and respiration. The NN model's greater emphasis on growing degree-days, in particular, suggests a stronger capability to integrate seasonal temperature dynamics and assess the cumulative impact of temperature on plant growth throughout the year.

Our results suggest that the NN model's decision process might align more closely with expected ecosystem carbon dynamics, while the RF model's predictions show a weaker alignment with underlying physical processes. This misalignment may





negatively impact the RF model's predictions under warmer RCP scenarios over longer time periods, as shown in some disparities between LPJ-GUESS outputs and RF predictions toward the end of the 21st century for carbon fluxes, for example.

## 5.3 Comparison to previous studies

A common approach in emulating spatially resolved ecosystem variables, particularly with statistical methods, involves developing multiple emulators tailored to specific plant functional types (PFTs), crop types, fire regimes, or biomes (Ahlström
et al., 2013; Ekholm et al., 2024; Franke et al., 2020). This strategy facilitates the approximation of complex ecological functions and has been effective for certain applications. However, we argue that biome-specific emulators may be less suited for modeling future climate scenarios, where biome shifts or changes in forest productivity due to $CO_2$ fertilization are expected to occur. Fitting the parameters of emulators to specific biomes also risks averaging out regional climate change effects, thereby reducing the model's ability to capture the nuanced interactions that drive ecosystem dynamics at finer scales.

While the development of an LPJ-GUESS emulator is not new, our approach differs from previously developed approaches. Eckholm et al. (2024) emulated the effects of climate change on C stocks as a linear function of global mean temperature changes and atmospheric $CO_2$ concentrations, with separate applications for each biome, while Ahlström et al. (2013) parameterized a statistical emulator mimicking the LPJ-GUESS results when forced by global temperature and atmospheric CO2 as sole drivers.  Although these approaches are computationally efficient and interpretable, their reliance on linear
regression may oversimplify the non-linear ecological responses to climate change and miss regional climate variations that differently impact biomes. In contrast, our approach proposes a single global emulator that does not rely on spatial proxies and it is not specific to a certain biome type or PFT. This biome-agnostic design allows the emulator to capture both global and regional climate dynamics without averaging the effects of climate change across biomes. This approach can also more effectively model biome shifts and other complex ecosystem responses to future climate scenarios.

## 5.4 Emulator application

The emulators developed in this study are lightweight models designed to simulate forest ecosystem carbon dynamics in response to climate change. While the emulators can be used directly for rapid simulations of carbon pools and fluxes without the need for solving an optimization problem, they were aimed at integration with the LandSyMM model. Future work will explore this integration to enable addressing questions related to the forest-based climate change mitigation potential in the
context of global environmental change. Within LandSyMM, the emulators will replace LPJ-GUESS simulations in an online coupling to optimize global forestry decisions over multi-decadal timescales. The emulators will provide annual forest productivity and track changes in C stocks and $CO_2$ emissions following clear-cuts. The land-use model within LandSyMM, PLUM, will provide information regarding the timing of clear-cuts and affected grid cells, so that the emulator can predict the target variables in the future based on the current state of C stocks, annual climate conditions and other relevant features.
Future development of the emulator may be required to accommodate additional forestry practices, such as various thinning



regimes, specific planted PFTs, or harvest probabilities. The emulator's flexibility means it can be retrained or fine-tuned for different horizontal resolutions, as well as adapted for alternative applications, such as more detailed management strategies.

## 6 Conclusions

The emulators developed in this study demonstrate the effectiveness of ML methods in accurately capturing the process-based dynamics of forest carbon stocks and fluxes under climate change scenarios. However, the differences in model performance and explainability highlight the trade-offs between the generalization ability and overall accuracy of each model type. The NN's tendency to over- and underestimate target variables is contrasted with its ability to generalize well to warmer climate scenarios by the end of the 21st century, and its decision making that is highly consistent with ecological processes.

Nevertheless, we do not discard the use of the RF emulator, particularly due to its overall higher prediction accuracy in the first years following a disturbance event. These findings emphasize the importance of model selection based on the specific task at hand and the trade-offs between accuracy and interpretability in future projections. The potential use of an ensemble model for emulation is also worth considering, as it could combine the strengths of both models while offering the advantage of faster predictions.

While the emulation development was non-trivial, the whole approach was developed using standard open source ML libraries, 470 facilitating replication and subsequent improvement efforts. The integration of such ML approaches into modeling frameworks has the potential to improve forest management optimization, offering a valuable tool for policy planning in the face of climate change.

## Code and data availability

The LPJ-GUESS version 4.1.1 model code used to generate data for emulation development is publicly available under the 475 Mozilla Public License 2.0 at the LPJ-GUESS community repository (Nord et al., 2021). Specific modifications made to the source code are detailed in Supporting Information S1 and in our GitHub repository: https://github.com/natel-c/lpjg-modif-emulator.

ISIMIP3b bias-adjusted atmospheric climate data used in our simulations was obtained from the ISIMIP Repository and is 480 provided under the CC0 1.0 Universal Public Domain Dedication. (Lange and Büchner, 2021)

All data, pre-trained models and pre-calculated SHAP values necessary to reproduce the figures are available under a Creative Commons Attribution 4.0 International (Natel de Moura et al., 2024) and can be accessed through the Zenodo repository at https://zenodo.org/records/14230951.




The code necessary to reproduce the figures are available under a Creative Commons Attribution 4.0 International (natel-c, 2024) and can be accessed through the Zenodo repository at https://zenodo.org/records/14231373.

Machine learning libraries used in this work include TensorFlow (TensorFlow Developers, 2024), Keras (Chollet, 2015),
scikit-learn (Pedregosa et al., 2011) and SHAP (Lundberg and Lee, 2017). Additional required libraries are specified in the environment files available in our GitHub repository.

**Author contribution**

AA and SR conceptualized the study. CN and AA secured funding. CN designed the experiments. CN and NH conducted the formal analysis and developed and evaluated the emulation approach described in this study. AA, DB, and PA contributed to
the analysis and discussion of the results. CN drafted the initial manuscript. CN, DB, PA, SR, and AA revised the manuscript. All authors approved the final version.

**Competing interests**

We declare that one of the coauthors is a member of the editorial board of Geoscientific Model Development.

**Acknowledgements**

The authors would like to thank the Alexander von Humboldt Foundation for providing the first author with a fellowship during the course of this research. During the preparation of this work the authors used AI assistants to improve the quality of the text, and to check grammar and spelling. After using these tools, the authors reviewed and edited the content as necessary and take full responsibility for the content of the publication.

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
