# Peer review of "Emulating grid-based forest carbon dynamics using machine learning: an LPJ-GUESS v4.1.1 application"

_EGUsphere, 2024_

## Author Comment (AC2)

Dear Thomas Oberleitner,

We appreciate your time and effort to provide comments on our preprint. Please see below our replies (in blue) to each of your points.
* * *
Thank you for this interesting preprint. Here are some notes and suggestions regarding the presentation of the results.

1. The preprint compares the predictive performance and parameter/response relationships of random forests and neural networks. The benefit of comparing two high-capacity/complexity models on tabular data is not clear, as it is self-evident that both can achieve good results given proper handling. In fact, because regularization in NNs can be more difficult, they are generally outperformed by easier to use off-the-shelf models such as random forest and gradient-boosting [1].

Furthermore, we don't know of any literature in ML research supporting the claim that NNs would be inherently better at intra -or extrapolation unless they incorporate domain specific properties [2], for example the network architectures used in physics-informed NNs. Nor do the results suggest systematically better generalization performance. In table 3, RF models show higher $R^2$ than NNs for all RCPs and the slightly better performance in table 4 could be due to the choice of hyperparameters, random seeds, etc.

We therefore suggest removing the model comparison and focusing on the RF model.

Thank you for this valuable feedback. We agree that RFs are often strong models for tabular data, and that both RFs and NNs demonstrated comparable predictive accuracy in our study. However, our objective here was to go beyond model accuracy and investigate explanatory power or model interpretability. In other words, we wanted to examine how different model architectures - with their different inductive biases - capture the true relationships in complex forest carbon dynamics. In this respect, we believe that our model comparison is a valuable contribution, especially given the scarcity of studies examining the trade-offs between model accuracy and interpretability in statistical models and emulators for ecological applications (Hu et al., 2023). We will clarify our rationale for comparing these models in a revised version, as this aspect could be better articulated in the objectives and discussion of the manuscript.

We also acknowledge your concerns regarding the claim that NNs are better at extrapolation, as this statement is not widely supported in the ML literature, and model performance can be highly dependent on the domain/data, and training process quality. Additionally, since our study does not include a systematic analysis to substantiate this claim, we will revise the manuscript to remove such statements.

2. The NRMSE in the model summary seems redundant as another scale-free metric in the form of the $R^2$ is provided. Additionally, NRMSE is highly sensitive to outliers, whereas $R^2$ is much less affected.

We included the NRMSE to provide an additional perspective on model performance, particularly for readers who may find error magnitudes more intuitive. Additionally, NRMSE facilitates comparisons with other machine learning emulators in the field, such as the work by Sun et al. (2022). However, if other reviewers strongly recommend its removal from the main text, we are open to relocating it to the supplementary materials

3. The point that the emulator reproduces LPJ-GUESS outputs well is made rather strongly in section 4. For example, line 215: "… emulators were able to generalize to LPJ-GUESS outputs produced with climate projections not included in the training data without a significant decline in performance". It is not clear what "significant" refers to, nor is this evident from table 3 and 4, which show low $R^2$ values for most responses. The next sentence highlights the low NRMSE values, which could be deflated due to outliers (see remark 2).

We suggest more careful wording regarding emulator performance and to put it into a more applied context, i.e., by highlighting its efficiency in a specific task. While it reproduces average outputs of LPJ-GUESS well (figures 2 and 4), it most probably cannot reproduce extreme outputs of the process model. An analysis of residuals can help in verifying that. The potential inaccuracies in predicting non-average responses should then be noted somewhere, as the emulator seems to be intended as a highly efficient proxy for the process model.

The statement in question (line 215) was intended to highlight the emulator's ability to generalize to unseen RCP scenarios. Specifically, when examining the $R^2$ values in Tables 3 and 4, we observe no decline or clear differences between the $R^2$ values for RCPs used during training (RCP2.6 and RCP8.5) and those for the unseen test scenarios (RCP4.5 and RCP7.0). We noticed that the proximity of this statement to the sentence on the emulator's overall ability to reproduce LPJ-GUESS outputs may lead to confusion, and we will revise the passage to improve clarity. We also appreciate your suggestion to analyse extreme model outputs, as this would provide important information into the emulator's performance. We plan to incorporate this analysis into a revised version of the manuscript.

Additionally, we will refine our wording to avoid overstating the emulators' ability to fully reproduce LPJ-GUESS outputs and to better contextualize its performance in their intended use.

4. The attribution of importance to features using Shapley values in the way it is presented could be misleading in the presence of correlations. This is a property of all data-driven models trained on correlated data, which is why all measures of importance are affected by this to varying degrees (e.g., total information gain in random forests, coefficients in linear models, etc.). In our experience, climate and other data used to train process model emulators are highly correlated and have a major effect on explanations. This can scramble the importance ranking of correlated features and even flip their Shapley value sign [3].

Furthermore, the text does not mention the ranking method for the features, which makes it hard to compare with the SHAP plots. Provided the authors stick to Shapley values, having the rank number included in the feature names in the plot would help to understand the conclusions drawn in the text.

We recommend supplementing correlation analysis, remove correlated features and/or weakening the language and inferences made about them. In many cases, feature selection algorithms can help in removing correlated features.

Alternatively, global explanations of feature importance could be used to rank features or supplement the Shapley results, such as contribution to loss function, global information gain, permutation importance, etc. As mentioned above, such measures are also not robust against correlations, but they might warp the results in a less drastic way. For some ML models, they are directly incorporated into feature selection algorithms [4].

Thank you for your comment and the references on this issue. We are currently studying the best way to address this limitation of the method. So far, we have come across and tested an approach described by Au et al. (2022) and Molnar (2022) for dealing with correlated features in this type of analysis. We plan to conduct a feature correlation analysis, group highly correlated features together, and then calculate and interpret Shapley values at the group level rather than for individual correlated features. While this approach is not perfect, as the interpretations are coarser at the group level, we believe it will make our feature importance interpretation more robust in light of the SHAP method limitations. We will also use Shapley-based global feature importance rankings and clustering analysis (as implemented in the Python SHAP library) to visually demonstrate feature redundancy in our importance plots. Any changes will be documented in the revised methods section.

Minor remarks
a. In the NRMSE equation (2), the term under the square root in the numerator should be divided by n.

We thank you for catching this error. We will correct Equation (2) in the revised manuscript.

References
[1]  V. Borisov, T. Leemann, K. Seßler, J. Haug, M. Pawelczyk, and G. Kasneci, "Deep Neural Networks and Tabular Data: A Survey," IEEE Trans. Neural Netw. Learn. Syst., vol. 35, no. 6, pp. 7499–7519, Jun. 2024, doi: 10.1109/TNNLS.2022.3229161.
[2]  V. N. Vapnik, "An overview of statistical learning theory," IEEE Trans. Neural Netw., vol. 10, no. 5, pp. 988–999, Sep. 1999, doi: 10.1109/72.788640.
[3]  K. Aas, M. Jullum, and A. Løland, "Explaining individual predictions when features are dependent: More accurate approximations to Shapley values," Artif. Intell., vol. 298, p. 103502, Sep. 2021, doi: 10.1016/j.artint.2021.103502.
[4]  "CatBoost Feature Selection." Accessed: Feb. 26, 2025. [Online]. Available: https://catboost.ai/docs/en/concepts/python-reference_catboost_select_features
* * *
Thank you once again for your constructive feedback. We look forward to implementing these points in a revised version of the manuscript.

Best regards,
Carolina Natel, on behalf of all co-authors

Hu, T., Zhang, X., Bohrer, G., Liu, Y., Zhou, Y., Martin, J., ... & Zhao, K. (2023). Crop yield prediction via explainable AI and interpretable machine learning: Dangers of black box models for evaluating climate change impacts on crop yield. *Agricultural and Forest Meteorology*, *336*, 109458.
Sun, Y., Goll, D. S., Huang, Y., Ciais, P., Wang, Y. P., Bastrikov, V., & Wang, Y. (2023). Machine learning for accelerating process-based computation of land biogeochemical cycles. *Global Change Biology*, *29*(11), 3221-3234.
Au, Quay, Julia Herbinger, Clemens Stachl, Bernd Bischl, and Giuseppe Casalicchio. "Grouped feature importance and combined features effect plot." *Data Mining and Knowledge Discovery* 36, no. 4 (2022): 1401-1450.
Molnar, Christoph. *Correlation Can Ruin Interpretability*. *Mindful Modeler*. Accessed 25 March 2025. https://open.substack.com/pub/mindfulmodeler/p/correlation-can-ruin-interpretability?utm_campaign=post&utm_medium=web

---

## Author Comment (AC3)

Dear Joe Melton,

Thank you for the time and effort you have taken to review our work. We believe that your constructive feedback will help us improve the presentation of our findings in a revised version of the manuscript.

We are responding to your remarks in detail below. Your comments were left in black text, our replies in blue, old passages in red, and new passages in green.

Natel and coauthors are interested in developing emulators to allow easier integration of ecosystem models (like LPJ-GUESS) in broader frameworks that couple multiple models together. LPJ-GUESS is sufficiently computationally expensive that an emulator could be valuable for the multimodel frameworks (in particular LandSyMM). They use two machine learning based approaches: random forests (RF) and neural network (NN). Both emulators were trained using LPJ-GUESS outputs for some historical and future simulations. The emulators differed in both their performance and the main variables they were sensitive to but both were much faster than LPJ-GUESS itself.

The paper is generally well written and easy to follow. The work falls well within GMD's area of interest. I think the work is suitable for publication but have several questions that I would like to see answered beforehand.

Main comments:

1. I liked that the authors used two different ML-based approaches in their emulators and then attempted to understand/interpret what each emulator was sensitive to. This is valuable information but it felt like only half the story. What was missing was what LPJ-GUESS is sensitive to. If the point of the emulators was to allow cheaper approximation of LPJ-GUESS (the 'model') then the most important thing is that the emulator is responding in the same manner and to the same variables as LPJ-GUESS. There are many plots showing trajectories of pools and fluxes for LPJ-GUESS and the emulators (e.g. Fig 2 and 4) but no similar plots showing sensitivity of LPJ-GUESS as there is of the emulators (e.g. Fig 6). I realize this is more challenging with LPJ-GUESS since it is inherently a completely different kind of model, but I struggle to understand how one can trust either emulator without knowing if it is actually mimicking the model's sensitivities (which under this circumstance has to be assumed to be perfect).

We agree that matching the emulators to the original model sensitivities is valuable. However, as the reviewer correctly points out, a direct comparison between the ML explainability analysis (Fig. 6 and Fig. 7: SHAP values) and the LPJ-GUESS sensitivity is challenging due to structural differences between the models. Nevertheless, we believe that interpreting and discussing the SHAP values, especially for features related to the LPJ-GUESS forcing (climate variables and atmospheric CO2 concentration), in light of previous sensitivity studies of LPJ-GUESS (Ahlström et al., 2013, 2017; Piao et al., 2013), could hopefully address the reviewer's comment and provide more insight into how the emulation approach compares to the original model sensitivities. Please note that we refrain from discussing the SHAP values alongside LPJ-GUESS *parameter* sensitivity analysis studies, as they are not directly comparable or relevant to the emulation approach presented here.

In response to a relevant comment from the community (CC1), we have also recomputed the SHAP values for groups of correlated features, and we have amended the section below to reflect these changes.

Old passage:

**4. Emulator explainability**

[…]

**4.2.1 Carbon stocks**

The RF and NN emulators exhibited varying feature importance and feature value impacts on carbon stock predictions (Fig. 6). For VegC, the NN model ranked pre-disturbance VegC pool, growing degree-days, and annual mean temperature as the most influential features, while the RF model ranked initial VegC, time since disturbance, and $CO_2$. There was a discrepancy between the models in terms of the impact of annual mean

temperature. While the NN model attributed a positive effect of lower temperatures on VegC, the RF model indicated a decrease in VegC under lower temperature values. Both models indicated a positive effect of the time since disturbance on VegC, which is consistent with the expected regrowth of vegetation carbon following a disturbance.

For SoilC, the NN model is primarily influenced by the initial SoilC state, temperature, and soil properties (sand, silt, and clay fractions), while the RF model relied mainly on the initial SoilC state. In NN model predictions, higher temperatures lead to an increase in soil carbon content. This result may seem counterintuitive, as higher temperatures usually increase Rh, reducing SoilC. However, in the warmer RCP scenarios where LPJ-GUESS simulates soil carbon increases, such as in tropical forests, NPP rises faster than Rh (Figs. 2 and 4). This might lead to a net carbon gain, suggesting that biomass input to the soil pool exceeds the increase in Rh.

For LitterC predictions, the NN model identified the initial LitterC state, growing degree-days, and soil sand fraction as the most significant features. In contrast, the RF model highlighted the initial carbon states of vegetation, soil, and litter, followed by time since disturbance and atmospheric $CO_2$ concentration, as more important. Both models concurred that in the initial years following a disturbance, lower time-since-disturbance values were associated with higher LitterC, which aligns with the observed peak in this pool after disturbance events.

Overall, variables associated with the carbon history of the stand before the disturbance are highly ranked in feature importance. Additionally, the time since disturbance, atmospheric $CO_2$ concentration, growing degree-days, and mean annual temperature also appeared to be crucial for predicting C stocks.

New passage:

4. Emulator explainability

[…]

For carbon stock predictions, the grouped SHAP analysis indicated similar feature importance rankings for both the RF and NN emulators, with only minor differences in the magnitude of feature group contributions to model outputs (Fig. 6). Precipitation and soil attributes emerged as the least influential factors, though these features exhibited slightly higher sensitivity in the NN emulators. Variables associated with the initial carbon state prior to disturbance emerged as the most influential across all carbon stock variables and emulators. Temporal features—such as atmospheric $CO_2$ concentration and time since disturbance—generally ranked second in importance, except for soil carbon predictions, where climate variables were the second most significant factor. Notably, for SoilC, variables beyond the initial carbon state contributed minimally to model outputs, in contrast to patterns observed for other carbon stock components.

For carbon flux predictions, the RF and NN emulators exhibited distinct feature importance patterns (Fig. 7). SHAP values from the RF emulator identified the initial ecosystem carbon state as the most influential feature group. In contrast, the NN emulator emphasized the joint contribution of atmospheric $CO_2$ concentration and time since disturbance for gross primary production (GPP), while climate variables were most influential for both net primary production (NPP) and heterotrophic respiration (Rh).

New figures:

[Figure]

*Figure 6: SHAP values for grouped features in carbon stock predictions, including vegetation carbon (VegC), soil carbon (SoilC), and litter carbon (LitterC), using (a–c) Random Forest (RF) and (d–f) Neural Network (NN) emulators. The Y-axis lists feature groups ranked by importance, where correlated features were grouped as follows: initial carbon (soilc_init, litterc_init, vegc_init), climate (temp, insol, temp_min, temp_max, mtemp_max, gdd0), and soil (clay, silt, sand). Precipitation (prec) was not correlated with other features. The X-axis displays SHAP values, which represent the impact of each feature group on model predictions.*

[Figure]

*Figure 7: SHAP values for grouped features in carbon flux predictions, including vegetation carbon (GPP), soil carbon (NPP), and litter carbon (Rh), using (a–c) Random Forest (RF) and (d–f) Neural Network (NN) emulators. The Y-axis lists feature groups ranked by importance, where correlated features were grouped as follows: initial carbon (soilc_init, litterc_init, vegc_init), climate (temp, insol, temp_min, temp_max, mtemp_max, gdd0), and soil (clay, silt, sand). Precipitation (prec) was not correlated with other features. The X-axis displays SHAP values, which represent the impact of each feature group on model predictions.*

The section below has been amended to discuss the SHAP values in the light of previous LPJ-GUESS sensitivity analysis studies. Please note that the text in this section has changed significantly, mainly due to newly calculated SHAP values for groups of correlated features, instead of individual features.

Old passage:

**5.2.1 Carbon stocks**

[revised manuscript text omitted]

Although we believe the above discussion demonstrates that the emulators reasonably capture LPJ-GUESS model sensitivities, we acknowledge that they do not fully replicate the entirety of LPJ-GUESS. This limitation arises from the fact that the emulator has been trained on a limited dataset.

However, we also argue that achieving complete sensitivity matching is neither strictly necessary nor always practical in emulation applications. Emulators are typically designed to approximate specific input-output relationships of (parts of) a complex model under defined conditions, rather than replicate every internal process or parameter sensitivity of the original model. To clarify this point further, we have added the following in Section 5.4 Emulator Application:

New passage:

It's important to note that while the emulators were generally able to reproduce LPJ-GUESS's outputs related to forest carbon dynamics for the employed RCP scenarios, they should not be expected to capture all original model sensitivities, including both parameter sensitivities (e.g. parameters governing vegetation dynamics), and the original model' physical responses (e.g. the response of carbon dynamics to atmospheric $CO_2$ outside the bounds of training data)".

2. I am quite skeptical of the claim that (L 342) 'NNs excel at modeling continuous relationships, making them more capable of generalizing to unseen data, particularly when extrapolation is required'. My read of Muckley et al. (2023) does not support this contention. Muckley et al. test out the performance of linear regressions and black box RF and NN models. For the interpolation tasks, the linear model was poor but when it came to extrapolation it could out perform the black box models in some of the tests (~40%). The authors state the '{linear regressions}... may be desirable over complex algorithms in many extrapolation problems because of their superior interpretability...'. Lakshminarayanan et al. (2017) nicely demonstrate this for a toy example using a NN whereby the NN extrapolates poorly (their Fig 1 - left panel is bounds of 5 NNs). They show that the uncertainty bound via an ensemble technique can be created that encompasses the true function. So, getting to my main concern, given that NNs do not extrapolate well (same with decision tree-based methods), how can we trust the NN/RF models when they are forced to extrapolate? The approach here doesn't have any way to uncertainty bound the emulator results so it can extrapolate (poorly) blindly to the user. I don't expect the authors to fix this problem right now, but I would like to see more discussion about this difficulty and how it could be addressed for emulators as their use if becoming more common.

Thank you for this valuable feedback. We fully acknowledge that our original statement about NN extrapolation was not accurate. Additionally, we have not conducted a systematic experiment in this study to test this claim, therefore we sincerely apologize for this oversight.

Our argument in that specific passage should not have been about *extrapolation*, but rather *interpolation*, and we explain why here. During our preliminary experiments, we observed that NNs were better interpolators in the neighbourhood of feature values (while still within the bounds of the training data, and therefore not *extrapolation*) compared to RFs. One possible explanation (among several) for this is the difference in inductive biases and function approximation capabilities inherent to each ML technique. For example, RFs are based on CART (classification and regression trees), which partition the data using step functions, resulting in a non-smooth relationship between features and target. In contrast, NNs can learn smoother functions. Our preliminary observations align with what is illustrated in Fig.1a and Fig.1b below (not from our work).

(a)                                                          (b)

[Figure]

Figure 1. Inductive biases effect on predictions. (a) Decision Tree vs. Random Forest, (b) Extrapolation in the feature space by different machine learning models. Source: Christoph Molnar (2025)

While changing the term *extrapolation* to *interpolation*, and a few more details in the text could, to some extent, improve the text to better reflect our view, we have opted to remove the entire paragraph. This decision is based on the fact that our experiments do not systematically test inductive biases, nor have we reported our initial

analysis in the manuscript. We believe that removing the passage below will ensure that the presentation of our findings is more accurate and better aligned with our experiments.

Old passage:

The generalization power of any learning algorithm depends on the inductive biases of that algorithm (Mitchell and Sheppard, 2019). RF models, for example, partition data through decision node splits using step functions that approximate relationships between inputs and outputs (Breiman et al., 2017). While this structure works well within the bounds of the training data, it limits the model's capacity to smoothly extrapolate outside the training data distribution. In contrast, NNs excel at modeling continuous relationships, making them more capable of generalizing to unseen data, particularly when extrapolation is required (Muckley et al., 2023). Given that our dataset contains less data for warmer climates near the end of the century, this limitation may affect the RF model's performance, particularly in extrapolating to these conditions. However, NNs are generally more robust in handling such extrapolation tasks due to their smoother function approximations. This suggests that RF may be more reliable when future projections remain close to the training data distribution or for interpolation tasks, while NNs could be better suited for scenarios where extrapolation is critical. This distinction is particularly relevant for climate assessments, where projecting across a wide range of future scenarios is essential for decision-making.

We have also added a paragraph in Section 5.4 to address the reviewer's comment on the extrapolation issue and uncertainty in ML emulations, as follows:

New passage:

Furthermore, ML-based emulators should not be assumed to reliably extrapolate beyond the training distribution without proper validation. In our study, both models were evaluated on scenarios that, while challenging, were within reasonable bounds of the training conditions. As demonstrated in Lakshminarayanan et al. (2017), NNs extrapolate poorly and uncertainty bounds using ensemble techniques may help to encompass the true function, an approach that could be explored in further development of emulation approaches.

Minor Comments:

Supplement - Can you explain more about the disturbance interval of 100 years and how that is applied? Also with fire off, the disturbance is then what? I see land use change is also not used.

We have now added the following paragraph to the Supplemental Materials to clarify this point in the section describing the LPJ-GUESS setup.

New passage:

In LPJ-GUESS, in addition to fire disturbances, we account for other external disturbances (e.g. windstorms, plant diseases etc) using a generic patch-destroying regime with a stochastic probability based on the expected return time. Disturbance return time varies substantially across the global forest area (Pugh et al., 2019), and the interval we have chosen is a simplification that has been adopted in several previous studies using LPJ-GUESS and other vegetation models, as reported by Zaehle et al., 2005.

L 84 - This sentence is a bit confusing as I found it less clear what the features were selected for.

This selection refers to the features used during emulation training. We have now revised the text as follows:

Old passage:

We selected 15 features, including variables related to climate, carbon states prior to a stand-replacing event, soil attributes, and a disturbance timer that tracks the time elapsed since the last stand-replacing disturbance (Table 1).

New passage:

We used 15 features as inputs to train the emulators. These included variables related to climate, carbon states prior to a stand-replacing event, soil attributes, and a disturbance timer that tracks the time elapsed since the last stand-replacing disturbance (Table 1).

L 112 - I wonder about the influence of this post-processing step for non-negative C stocks. How often did this come up? Were the instances where this came up for regions with very low stocks such that the inaccuracy would be small? (e.g. true value is 0.1 kg C/m2, so going negative is fairly reasonable but if it was really supposed to be 10 kg C/m2 then that is a big problem). This also demonstrates a problem with using an off-the-shelf NN whereby it has no knowledge of boundaries that one like a physics-informed ML model could.

The post-processing step was included as a safeguard against unrealistic values, which can arise when training NNs. However, we had not previously assessed its impact on our predictions. Thank you for raising this point. Upon analysis, we found that negative predictions occurred in only 0.09% of cases in our test dataset, primarily affecting vegetation carbon estimates. These errors were generally small, with a mean absolute error of 0.78 kg C/m² and a maximum absolute error of 2.89 kg C/m². Notably, 90% of these cases occurred when the true values were below 1.5 kg C/m². We will include the additional scripts for this analysis in our code repository, in a subfolder called tests/.

L 137 - sampled by grid cell, time, or ?

The random sampling was across grid cells and time steps. We have revised the text to clarify this as below:

Old passage:

To reduce computational time, the SHAP analysis was conducted on a randomly sampled subset of the test dataset (n=500) from the predictions made using the MPI-ESM1-2-HR forcing data for the historical period and climate scenarios (RCP2.6, RCP4.5, RCP7.0 and RCP8.5).

New passage:

To reduce computational time, the SHAP analysis was conducted on a randomly sampled subset of the test dataset (n=500), drawn across grid cells and time steps from the predictions made using the MPI-ESM1-2-HR forcing data for the historical period and climate scenarios (RCP2.6, RCP4.5, RCP7.0, and RCP8.5).

L 153 - with the meteorology/climate of MPI..., not the actual climate model itself.

Correct. Thank you! We have fixed this as below:

Old passage:

For our benchmark, we estimated the computational gain using predictions for a 165-year historical period (1850–2015) simulated with the MPI-ESM1-2-HR climate model.

New passage:

For our benchmark, we estimated the computational gain using predictions for a 165-year historical period (1850–2015) simulated with the climate of MPI-ESM1-2-HR climate model.

L 155 - Does LPJ-GUESS not have a dependence for computational cost on the number of PFTs present or the soil permeable depth?

Yes, indeed, if multiple woody PFTs can establish in a grid cell, each may form a cohort (a group of trees of the same PFT). More cohorts mean more computations (e.g., growth, competition, mortality), increasing runtime. The computational time is also dependent on the number of layers and soil permeable depth, which, in LPJ-GUESS, consists of 15 soil layers (each 10 cm deep) plus 5 additional layers for temperature padding.

Acknowledging that runtime may vary across grid cells due to these factors, we have now recomputed the computational gain by using a representative set of global grid cells (the same used for the validation period, n=344, Fig.1 of the manuscript). The recalculated value is presented in the revised text below.

Old passage 1:

We used a single grid cell (0.5° x 0.5°) for this comparison, noting that the gain would scale proportionally for larger areas.

New passage 1:

We used 344 grid cells from the validation set for this comparison, expecting that this representative set would capture runtime variations due to differences in the number of simulated woody PFTs and soil permeable depth across grid cells.

Old passage 2:

The emulators demonstrated a significant reduction in simulation execution time, with a 97% decrease observed when compared to the execution time of LPJ-GUESS.

New passage 2:

The emulators demonstrated a significant reduction in simulation execution time, with a 99% decrease observed compared to the execution time of LPJ-GUESS. The LPJ-GUESS runtime for the validation set was 5765.56 seconds, whereas the RF emulator ran in 1.26 seconds and the NN emulator in 2.77 seconds on a single-processor computer. However, since LPJ-GUESS grid cell simulations are typically run in parallel on high-performance computing systems, we also calculated the computational gains per grid cell by dividing the total runtime by 344 (number of grid cells used in this analysis) for both the original model and the emulators, resulting in an average 95% decrease in runtime with the emulators.

L 158 - If it takes 5000 sims to train the emulator but actually running the model that uses the emulator only happens 1000 times then you may end up with no net benefit. Also, sorry if I missed it, how many simulations did you need to train the emulator?

The emulator training required 25 simulations (4 RCPs + historical period x 5 GCMs) for each of the 3448 grid cells used in training, validation and test, resulting in a total of 86,200 samples, which is significantly fewer than the 574,650 predictions needed to run the full LPJ-GUESS model for all forested grid cells (n= 23,007), climate models and scenarios. While this upfront cost might seem substantial, the real benefit of the emulator lies in its integration with the LandSyMM model. Within this framework, the emulator needs to be called repeatedly to evaluate a wide range of management scenarios to supply e.g. timber demand or carbon sequestration options, which would be computationally infeasible using the full LPJ-GUESS model. Therefore, we are positive that the emulator will, in the long run, far outweigh the initial training cost when integrated into LandSyMM. The number of samples needed to develop the emulator is reported in Table 2.

L 207 - Could you give the real values in addition to the percent. It would be nice to see how much in clock time these cost (acknowledging it is system dependent).

Of course, please see the revised passage above with the recalculated computational gain, where we mention the clock times.

L 215 - I think the lack of decline in performance is simply due to training with the most extreme ends of the scenarios. This ensured that you were interpolating as much as possible. This is likely the only reasonable approach given the these techniques do not extrapolate well (see one of my main comments). But it means that the emulator always requires retraining for new scenarios and the scenarios always need be more extreme than what the actual system should realistically experience. I think some aspects of this bear mentioning.

Definitely, we appreciate your comment and are addressing this along with your main comment to clarify that extrapolation is a limitation of black-box models. The revised paragraph is presented below:

Old passage:

As shown in Table 3, the emulators were able to generalize to LPJ-GUESS outputs produced with climate projections not included in the training data without a significant decline in performance.

New passage:

As shown in Table 3, the emulators were able to generalize to LPJ-GUESS outputs generated with climate projections (RCP4.5 and RCP7.0) that were not included in the training data, without a significant decline in performance compared to the training scenarios (RCP2.6 and RCP8.5). This indicates that the emulators can generalize across intermediate emission scenarios, which fall within the range defined by the low (RCP2.6) and high (RCP8.5) extremes used during training. However, extrapolation beyond this range would require additional training and evaluation, as black-box models are not inherently robust in extrapolation tasks (Muckley et al., 2023).

L 218 - 'greatest accuracy' - by NN? Unclear as written.

Reply: No, we are actually referring to overall accuracy (for both NNs and RFs) compared to other targets. We have revised the sentence for clarity:

Old passage:

Among the carbon stock variables, SoilC was predicted with the greatest accuracy, exhibiting the lowest error and highest $R^2$ values.

New passage:

Among the carbon stock variables, SoilC was predicted with the highest accuracy by both NNs and RFs, showing the lowest error and highest $R^2$ values.

Fig 2 and 4 - What about adding new plots presenting these and the fluxes as cumulative plots so the impact of over/under predicting over time are visible? The fluxes is important as it has impact on how much C the land surface takes up/releases. The stocks as it changes how much C is emitted during disturbance or land use change. A cumulative plot can show the effect across the simulated period.

Reply: We have now generated the suggested plots (shown below) and will include them in the Supplemental Materials due to space constraints in the main manuscript. These plots illustrate cumulative biome-specific carbon (stock/flux) changes (1900–1930 vs. 2070–2100) across grid cells and time for each scenario, based on the test dataset. If this interpretation does not fully align with your suggestion, we are happy to refine the analysis further, please let us know any additional specifics you'd like included.

[Figure]

*Figure 2. Biome-specific change in carbon stocks (1900 – 1930 to 2070 – 2100) for the test set. Values represent the cumulative stocks across time and grid cells for a range of climate change scenarios.*

[Figure]

*Figure 3. Biome-specific change in carbon fluxes (1900 – 1930 to 2070 – 2100) for the test set. Values represent the cumulative fluxes across time and grid cells for a range of climate change scenarios.*

*Lakshminarayanan, B., Pritzel, A., and Blundell, C.: Simple and scalable predictive uncertainty estimation using deep ensembles, arXiv [stat.ML],31st Conference on Neural Information Processing Systems (NIPS 2017), Long Beach, CA, USA. https://arxiv.org/pdf/1612.01474*

Once again, we sincerely appreciate the reviewer's feedback, which will help strengthen the manuscript.

Best regards,

Carolina Natel, on behalf of all coauthors

---

## Author Comment (AC4)

Dear Reviewer,

We appreciate your time and effort in reviewing our work. Below, we provide our replies (in blue) to your comments. For easier reference, we've highlighted the revised text in green.

Carolina Natel et al. developed both a Random Forest model and a neural network model to emulate the dynamics of ecosystem carbon fluxes and carbon pool changes. These machine learning models have been widely applied to land models or components in the past and have consistently demonstrated effectiveness. Similarly, this study shows reasonable performance in emulating the target variables. The paper is well-written, with well-documented data and code. Overall, this is a solid modeling paper. Below, I have a few specific comments:

1. Interpretability vs. Physical Consistency

My primary concern is the balance between model interpretability and physical consistency. While SHAP has been used to interpret the ML models, it does not ensure that the emulators capture established physical knowledge embedded within the land model (in this case, LPJ-GUESS). I encourage the authors to further explore this aspect, as it is fundamentally important to understand the functional relationship emerging from ML emulators.

For example, in land models, atmospheric $CO_2$ concentration is a key driver of vegetation productivity, while temperature (T) strongly influences soil carbon stocks. Ideally, such first-order relationships should also be reflected in the trained ML emulators. One way to test this would be to leverage factorial LPJ-GUESS simulations with:

(a) Future SSP climate scenarios + historical atmospheric CO2 levels.

(b) Future SSP CO2 levels + historical climate conditions (e.g., repeated climate from 2010–2020).

If the trained ML emulators can reproduce the results of these factorial runs, it would provide strong evidence that the emulators have captured critical relationships between environmental drivers and carbon dynamics.

We appreciate this comment and recognize the importance of evaluating whether the emulators capture the physical relationships embedded in LPJ-GUESS, such as those mentioned by the reviewer. In response to the first comment from another referee (RC1), we have revised our discussion section on ML explainability, specifically interpreting the SHAP values in the context of previously published LPJ-GUESS sensitivity studies. We invite the reviewer to refer to the updated section, which can be found in our reply to RC1 (https://doi.org/10.5194/egusphere-2024-4064-AC3).

Regarding the suggestion to evaluate the emulation using factorial LPJ-GUESS simulations, we offer the following clarifications. A similar approach was partially explored during the early stages of emulator development, as outlined in the manuscript (Lines 355–360). Our initial hypothesis was that training the emulators on a factorial experiment, with independent variations in temperature, precipitation, and $CO_2$, would allow them to generalize to future RCP scenarios. However, we encountered a key challenge the factorial experiments generated feature spaces with physically implausible combinations in the feature and target space (e.g., low $CO_2$ paired with high temperatures, which only arise in extreme future scenarios with elevated $CO_2$, or extreme unrealistic ecosystem carbon in regions in which bioclimatic or soil constraints would not allow). This mismatch between training data and the intended application (RCP-based projections) led to poor generalization, as ML models rely on statistical patterns within the training distribution.

To address this issue, we tested a two-step training approach using neural networks. We pre-trained the NN on factorial experiments and fine-tuned it on RCP scenarios. However, the extreme imbalance between these two datasets caused one of two problems: (i) models became biased toward the factorial scenarios and performed poorly on realistic RCP trajectories, or (ii) when the weights were allowed to adapt flexibly to the new training data, fine-tuning effectively overwrote the pre-trained weights. This rendered pretraining ineffective and substantially increased training time.

It is important to note that purely ML-based emulators do not explicitly enforce process-based relationships in the same way as process-based models. As such, they differ fundamentally from process-based models and

cannot generalize across different data distributions. We could have included in our training process some sort of pre-processing technique to balance the training data, in which we for example, oversample the examples similar to realistic RCP scenarios to balance out the training data, however, having such a flexible emulation approach was out of the scope of this work. Despite this limitation, our ML emulators remain highly effective for their intended application. Our training dataset consisted of physically plausible RCP scenarios at each grid cell, ensuring that the emulators learned realistic covariances among drivers. We do, however, acknowledge that further development of the LPJ-GUESS emulator for LandSyMM could benefit from constraining the emulator with LPJ-GUESS 'physics', which would make it more useful in a broader range of applications. We believe the revisions we made in response to RC1 (Main Comment #1) partially address this concern, especially the newly added paragraph in Section 5.4, "Emulator Application", which reads:

New passage to address RC1 and RC2:

It's important to note that while the emulators were generally able to reproduce LPJ-GUESS's outputs related to forest carbon dynamics for the employed RCP scenarios, they should not be expected to capture all original model sensitivities, including both parameter sensitivities (e.g. parameters governing vegetation dynamics), and the original model' physical responses (e.g. the response of carbon dynamics to atmospheric $CO_2$ outside the bounds of training data).

Additionally, we plan to expand the explanation of our preliminary tests in Lines 355–360 to further clarify the issues encountered when testing the emulator with factorial analysis scenarios, as discussed in this reply.

2. Justification for Annual Time Step

Further justification is needed regarding the choice of an annual time step. Land models typically operate at much finer temporal resolutions (e.g., daily, hourly, or at least monthly). It would be helpful to explain why the annual scale was selected and how potential loss of information at shorter timescales may affect the emulator's performance.

Yes, indeed, some LPJ-GUESS processes are updated daily (e.g., phenology and soil organic matter dynamics), while carbon allocation occurs annually. Our decision to use an annual time step for the input features was based on practical considerations. On one hand, we aimed to develop an emulator that could replicate LPJ-GUESS's interannual variability under RCP scenarios with sufficient accuracy (Fig. 2 and Fig. 4). On the other hand, we wanted to avoid including variables that might introduce unnecessary complexity without improving model performance. While finer temporal resolutions capture more information, they also introduce significant noise and variability, which may not enhance emulator performance and could even degrade it, in addition to increasing model complexity and training time.

We recognize that certain intra-annual climate information is essential for modeling the impact of climate change on carbon dynamics, such as the effect of rising mean annual temperatures on extending the growing season and increasing GPP, particularly in boreal regions (Piao et al., 2012). Therefore, we included seasonality-related variables, such as the highest mean monthly temperature (m_temp_max) and total annual growing degree days (gdd0), to capture these climate-driven processes while maintaining model simplicity.

The following clarification has been added to the methods section:

New passage 1:

To emulate the LPJ-GUESS model, we used input features aggregated to an annual time step. While some LPJ-GUESS processes operate at finer temporal resolutions (e.g., daily updates for phenology and soil dynamics), key carbon fluxes, such as allocation, are computed annually. Our goal was to develop an emulator that accurately captures interannual variability in carbon dynamics under future climate scenarios while avoiding the complexity and noise associated with higher-frequency inputs. To retain essential intra-annual climate signals relevant to carbon responses, such as the effects of seasonality on productivity, we included variables such as the total annual growing degree days above 0°C (gdd0). This approach balances model simplicity with the need to represent critical climate-driven processes affecting carbon dynamics.

Additionally, we have included the following statement in the discussion of model explainability to highlight the potential loss of information at shorter timescales. This sentence will be included after the discussion of the SHAP values alongside LPJ-GUESS climate sensitivity.

New passage 2:

It is important to note that, due to the use of an annual time step for input features, some LPJ-GUESS sensitivities to intra-annual climate cannot be fully captured in our simulations. However, we believe the selected input features are well-suited for capturing the interannual variability in carbon dynamics for our intended application.

3. Capturing Inter-Annual Variability

Given the focus on annual time steps, evaluating the emulator's ability to capture inter-annual variability in carbon fluxes (in addition to long-term trends) would be an important validation metric. Although the training is performed at the grid cell level (random sampling), it may also be valuable to include spatially aggregated fluxes (e.g., global/regional totals) as part of the loss function. This could improve the model's ability to represent inter-annual variability at regional or global scales.

As we have demonstrated, the emulator not only captures long-term carbon dynamics trends (e.g., long-term carbon uptake or losses), but also the year-to-year variability. Therefore, we are uncertain about the reviewer's comment. It's possible the reviewer is referring to intra-annual (monthly or seasonal) variability in carbon dynamics; however, this does not align with the intended use of the emulator, nor is it a standard output in LPJ-GUESS climate assessment studies.

The suggestion to include spatially aggregated fluxes and more physics in the loss function is indeed appreciated, and we might consider improving the realism of the emulators along with further applications we might need in the future, for example, representing a diverse range of forest management options (thinning, planted species etc).

4. Treatment of Disturbance Intensity

The results show disturbance as one of the most important features. However, it remains unclear how disturbance intensity (e.g., fractional area burned by wildfire, or land-use/land-cover change) is handled. How does the ML model represent partially disturbed grid cells? Additional clarification on this point would be needed.

To clarify, the "disturbance" feature refers to the time since the last stand-replacing disturbance, such as a forest clearcut event in the LandSyMM framework. This feature is not designed to capture detailed disturbance intensity, but rather to track forest recovery. To avoid any confusion, we are adding a sentence in our Methods section to clarify what type of disturbances are being represented in the emulation and what this feature means.

New passage:

In our LPJ-GUESS setup, we simulate only Potential Natural Vegetation and do not account for land-use changes. Additionally, since the emulator is designed to predict forest carbon potentials, we have disabled the fire module. The only disturbance incorporated in the emulation is the LPJ-GUESS representation of external disturbances (e.g., windstorms, plant diseases) through a generic patch-destroying regime with a stochastic probability based on the expected return time. Disturbance return times vary significantly across global forest areas (Pugh et al., 2019), and the interval chosen in this study (100 years) is a simplification commonly adopted in previous studies using LPJ-GUESS and other vegetation models, as reported by Zaehle et al. (2005).

As our emulation application is designed to model forest regrowth following a clearcut event within LandSyMM, we have included a feature called "time since the last disturbance" (in years) to track forest recovery. Within LandSyMM, this feature is reset to zero whenever a clearcut is performed.

Additionally, we believe there was a misunderstanding regarding the emulator's application, and we apologize for not making this clearer earlier. The LPJ-GUESS emulator will serve as a fast proxy for carbon dynamics when coupled with land-use models (e.g., PLUM within LandSyMM) to optimize future decisions. However, it

is not intended to fully replace LPJ-GUESS within the LandSyMM framework. For example, the full LPJ-GUESS model will still be used to simulate crop productivity and other land uses that do not involve forests. Once the forest use scenarios are optimized for a certain simulation step, the resulting maps will be passed back to the original LPJ-GUESS model to predict actual carbon dynamics, including full disturbance effects (e.g., we might activate the fire module depending on the scenario). This final simulation step will ensure that we fully utilize the original model's sensitivities, but it's not computationally intensive.

To solve this misunderstanding, we are replacing the old passage:

Within LandSyMM, the emulators will replace LPJ-GUESS simulations in an online coupling to optimize global forestry decisions over multi-decadal timescales.

New passage:

In the context of the LandSyMM application, where we couple a land-use predicting model (e.g., PLUM) with LPJ-GUESS, the emulator will replace the original model at each coupling time step (e.g., every 5 years) to quickly estimate forest carbon potentials. This eliminates the need for full 100-year forest potential simulations for several scenarios. However, the emulator will not be used to predict actual vegetation carbon under the predicted land use for the next 5 years. Instead, a real LPJ-GUESS simulation will estimate vegetation carbon based on the predicted land use, and a new coupling step will then occur.

5. Recommendations for future work: Since land models simulate continuous, time-dependent changes in carbon fluxes and pools, it may be worthwhile to explore time-series ML models (e.g., RNNs, LSTMs, or Transformers) in future work. Such models could potentially outperform static models like Random Forests and ANNs by better capturing temporal dependencies and dynamics.

Thank you for this suggestion. We agree that exploring time-series ML models, such as RNNs, LSTMs, or Transformers, could be highly beneficial as we expand the emulator's capabilities to include more complex dynamics, such as multi-year recovery after disturbances and memory effects from land-use changes in soil carbon.

Thank you once again for your valuable feedback. We hope that we have addressed your concerns and we are positive that your comments will help strengthen our manuscript.

Best regards,

Carolina Natel, on behalf of all coauthors

---

## Author Response (AR1)

Dear Editorial team,

We thank you for considering our revised manuscript for publication, and the reviewers for their constructive feedback, which has helped us improve the manuscript.

Please find below a summary of the main revisions made to the manuscript in response to the comments and suggestions received during the Open Discussion phase. A detailed point-by-point response to all comments (RC1 and RC2) was previously uploaded as a PDF during the discussion phase (https://doi.org/10.5194/egusphere-2024-4064-AC3 and https://doi.org/10.5194/egusphere-2024-4064-AC4)

**Summary of Main Revisions:**

1. Recomputed the SHAP values analysis using grouped features, addressing the suggestion from CC1.

2. Revised the Emulator Explainability results and discussion sections to reflect the updated SHAP analysis and and the interpretations introduced in point 3.

3. Integrated suggestions from RC1 and RC2 by interpreting SHAP values in the context of previously published LPJ-GUESS model sensitivity studies.

4. Added clarification on the limitations of the emulation framework in Section 5.4 (Emulator Application), in response to RC1's comments.

5. Removed the statement regarding neural network extrapolation and added a note in Section 5.4 to address related comments from both CC1 and RC1.

6. Implemented several minor corrections and clarifications based on feedback from RC1 and CC1.

7. Addressed the technical editor's suggestions (CE1) concerning the code and data availability statement.

**Items Not Changed:**

1. We did not include a residual analysis, as suggested by CC1. We believe that the combination of three evaluation metrics (NRMSE, $R^2$, and Relative Bias), along with trajectory plots and spatial error maps, provides a sufficiently comprehensive assessment of emulator performance for the intended application. Moreover, adding an additional analysis would significantly increase the number of figures and potentially exceed the manuscript's word and space limits.

2. Regarding the use of factorial LPJ-GUESS simulations to evaluate emulation performance: as explained in detail in our response to RC2, this approach is not viable for our framework. Additional justification has been added to the manuscript in the Emulator Performance section.

We hope the revised version meets the expectations for publication.

Best regards,

Carolina Natel de Moura, on behalf of all co-authors.